behaviour, ecology

dynamic illumination, foraging, motion camouflage, signal masking, water caustics

**Author for correspondence:**
S. R. Matchette
e-mail: sam.matchette@bristol.ac.uk

# Underwater caustics disrupt prey detection by a reef fish

S. R. Matchette[1,2], I. C. Cuthill[1], K. L. Cheney[3,4], N. J. Marshall[3] and N. E. Scott-Samuel[2]

[1]School of Biological Sciences, University of Bristol, Tyndall Avenue, Bristol BS8 1TQ, UK
[2]School of Psychological Science, University of Bristol, Woodland Road, Bristol BS8 1TN, UK
[3]Queensland Brain Institute, and [4]School of Biological Sciences, University of Queensland, Brisbane, Queensland 4072, Australia

 SRM, 0000-0003-4503-8275; ICC, 0000-0002-5007-8856; KLC, 0000-0001-5622-9494; NJM, 0000-0001-9006-6713

Natural habitats contain dynamic elements, such as varying local illumination. Can such features mitigate the salience of organism movement? Dynamic illumination is particularly prevalent in coral reefs, where patterns known as 'water caustics' play chaotically in the shallows. In behavioural experiments with a wild-caught reef fish, the Picasso triggerfish (*Rhinecanthus aculeatus*), we demonstrate that the presence of dynamic water caustics negatively affects the detection of moving prey items, as measured by attack latency, relative to static water caustic controls. Manipulating two further features of water caustics (sharpness and scale) implies that the masking effect should be most effective in shallow water: scenes with fine scale and sharp water caustics induce the longest attack latencies. Due to the direct impact upon foraging efficiency, we expect the presence of dynamic water caustics to influence decisions about habitat choice and foraging by wild prey and predators.

## 1. Introduction

Variation in local illumination (hereafter, 'dynamic illumination') is an important visual component of most habitats [1–3], but is particularly pertinent if the variation is rapid, such as that elicited by water caustics. Water caustics (or 'wave-induced flicker') are an optical phenomenon created by the refraction and convergence of light rays through the curvature of a water surface [4–9], visible only when projected upon a reflective surface or scattering media (e.g. object, substrate, suspended particles [3]). When the water surface is disturbed by wind and waves, the light intensity at any specific location flickers over time, creating dynamic visual 'noise'. This intensity variation potentially makes objects harder to detect, and has even been linked to the initial evolution of colour vision itself, because the ratio of different wavelengths is more stable than the absolute intensity of light reflected from an object [10]. Numerous abiotic factors govern the form of water caustics, including the strength and direction of wind, the solar altitude angle, the turbidity of the water and the depth of the substrate [6]. However, the effect of water caustics on the visual performance of aquatic animals has received little attention.

When water caustics are projected upon a three-dimensional object, that object's shape and orientation determine the form of water caustics [6]. Projections onto planes parallel to the water surface comprise a mosaic of low-intensity, polygonal patches (hereafter, 'caustic shade') that are irregularly enclosed by high-intensity light (hereafter, 'caustic boundaries'), while perpendicular surfaces elongate the mosaic into linear bands [6] (figure 1*a*). Indeed, it has been suggested that some markings of pelagic fish, such as vertical barring, match the elongate form of water caustics for camouflage [6,11].

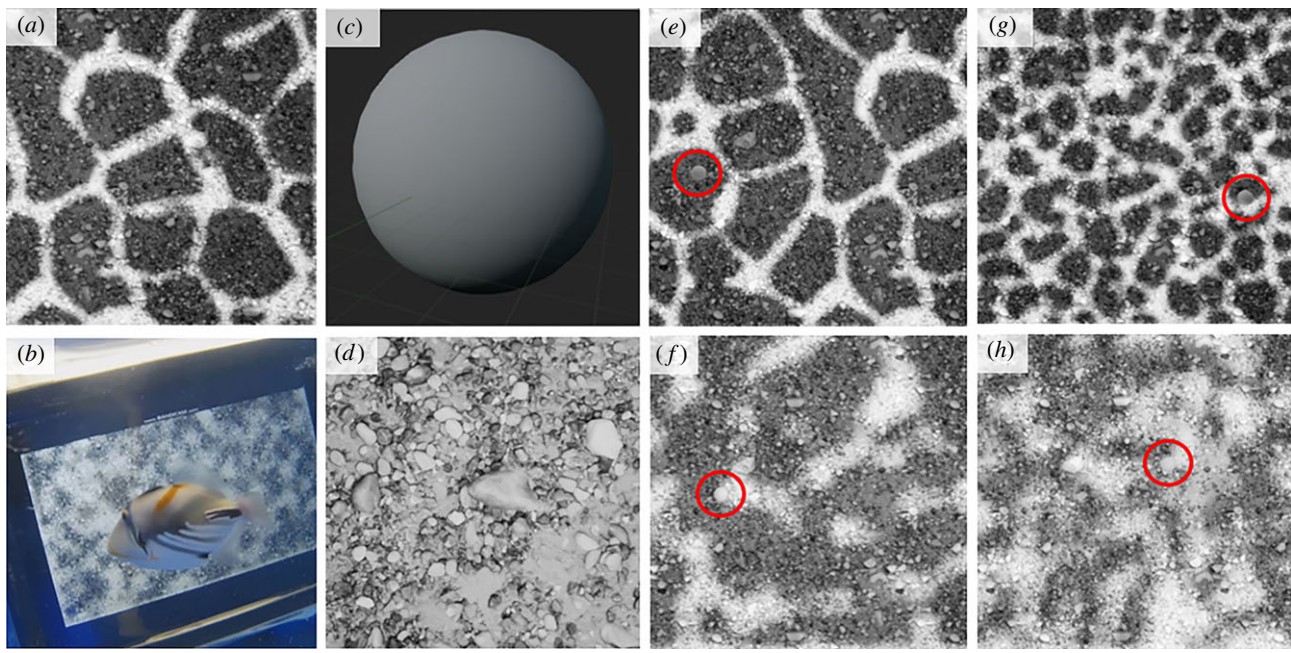

**Figure 1.** (*a*) A screenshot highlighting the visual features of water caustics: regions of low-intensity light ('caustic shade') irregularly enclosed by high-intensity light ('caustic boundaries'). (*b*) A frame taken from the *post hoc* video analysis showing an individual Picasso triggerfish, *Rhinecanthus aculeatus*, searching the iPad screen for the prey item. (*c*) A close-up of the prey item outside the experimental context. (*d*) Screenshot of the tiled river pebbles image used as experimental backdrop for all experimental trials. (*e–h*) Screenshots of an experimental trial from each non-dynamic treatment group with the prey item (circled) midway through moving across screen. Movement occurred from the furthest region on the left to the furthest on the right (or vice versa) at a constant speed of 24 mm s$^{-1}$ (6.9 deg s$^{-1}$). Caustic shade could be either coarse (*e,f*) or fine (*g,h*) scale, while caustic boundaries could be sharp (*e,g*) or diffuse (*f,h*). (Online version in colour.)

There is also growing evidence, in relation to camouflage and concealment, to suggest that the visual noise of dynamic illumination, such as water caustics, could reduce the costs of organism movement (see [12–15]). This is analogous to the movement of background objects (e.g. windblown vegetation), already known to affect both camouflage [13,16–18] and signalling [19–22]. This concept can be understood within a unifying framework of a signal-to-noise ratio [23], whereby motion signals go undetected if they fall within the distribution of motion noise. Matchette *et al*. [24] have highlighted the impact of dynamic illumination using computer-simulated scenes: human participants were significantly slower and more error-prone when detecting moving prey within scenes that contained dynamic illumination (dappled light and water caustics). While these findings are informative, we still know very little about how water caustics (and the subsequent visual noise) may influence the perception and behaviour of wild organisms.

Using a similar paradigm to that of Matchette *et al*. [24], we explore whether prey detection by a visually guided predator will be similarly disrupted by dynamic illumination, hypothesizing that water caustic flicker will mask the motion of a target prey item. Individual Picasso triggerfish (*R. aculeatus*; family: Balistidae) were trained to find and attack a moving prey item within simulated scenes with varying forms of water caustics (figure 1*b*). Picasso triggerfish have a broad distribution in shallow marine environments across the Indo-Pacific region, typically associating with coral reefs or rubble [25], and feed upon a variety of (primarily) benthic organisms [26] (see electronic supplementary material, movie S1). Moreover, the visual system of Picasso triggerfish has been extensively studied [27–30]. We use scenes with static caustics as control treatments that, while non-existent in nature, provide an opportunity to isolate illuminant motion and control for spatial complexity, as the latter has been

shown to reduce search efficiency in some taxa [23,31,32]. We also begin to explore whether the disruptive effect of dynamic illumination is relative to specific visual features of water caustics (beyond general motion), by manipulating two features of water caustics: the scale of caustic shade and the sharpness of caustic boundary, both of which are broadly associated with a change in depth [6]. With signal-to-noise ratio (SNR) in mind, we hypothesize that water caustics with sharp boundaries and fine shade scale—which represent those most acute in shallower waters—will induce the greatest attack latencies upon the current prey item.

## 2. Material and methods

### (a) Animals
A total of sixteen wild Picasso triggerfish were caught using hand nets and clove oil from shallow reef flats off Casuarina Beach, Lizard Island, Great Barrier Reef, Australia (14°40′8″ S, 145°27′34″ E) and released at the same location once the study was completed. Fish were collected under a Great Barrier Reef Marine Park Authority Permit G16/38497 and Queensland General Fisheries Permit 183990. All procedures were approved by the Animal Welfare and Ethical Review Body of the University of Bristol (UIN/UB/18/084) and the Animal Ethics Committee at the University of Queensland (QBI/304/16). Fish were measured upon capture and ranged from 65 to 130 mm (standard length; excludes caudal fin): individuals were deemed to be subadults and adults, and displayed similar levels of motivation to peck at prey items of the size presented.

Fish were housed individually in experimental aquaria (blue plastic tanks; 68 l volume; 650 mm × 410 mm × 395 mm) exposed to ambient daylight. Shade nets were fitted around the workbench to reduce the impact of direct sunlight during the early morning and late afternoon hours. Each aquarium had a seawater inlet

(from the source of capture), an outlet pipe and an appropriately sized shelter. All tanks were labelled with the date and location of capture, individual ID and number. An acclimatization period of 24 h was permitted before beginning any feeding regime. Fish were initially fed thrice daily (morning, noon and afternoon) to habituate to human presence and introduce their reward food item: a small (2 mm) piece of diced squid (*Doryteuthis opalescens*; Qualy-Pak Inc., CA, USA) which was offered with tweezers or, if not eaten directly, dropped to await later consumption (no longer than 30 min). Fish only began training once they consistently and readily took food directly from the tweezers.

## (b) Stimuli scene generation

While some terminology in this section may appear ambiguous, it derives directly from the software UNREAL ENGINE 4 (Epic Games, www.unrealengine.com) and therefore, to maintain clarity and replicability, we have chosen the retain the same terminology. Simulated scenes and prey items were constructed in UNREAL ENGINE 4 in an identical manner to Matchette *et al.* [24]. There were seven key components that formed the core of the experimental zone: (from bottom up) floor, spawn areas, prey item, camera item, water surface plane and the lighting systems. Each had particular settings ('blueprints') associated, which could be coded in various ways to alter performance and behaviour.

The floor component was a plane (static mesh) coated in a default material acquired from the free demonstration asset package, 'Kite Demo'. A material was attached to the floor component that would provide a background to the trials in all experiments. Backgrounds comprised an image, sourced from the software's default asset package, that represented river pebbles ('M_Tile_RiverPebbles'). We used the selected background 'out of the box', with range and mean of RGB values as supplied by UNREAL ENGINE, as these were already judged to be realistic. The background had no single dominant spatial frequency (i.e. no predominant pebble size), with a log–log plot of the amplitude of the spatial frequency against the frequency itself (down to a frequency equivalent to half the diameter of the target) having a slope of −0.91. Such a relationship is typical for many natural scenes, a slope of −1 being common [33]. The amplitude–frequency relationship for objects smaller than about half the size of the target was much steeper. That is, higher spatial frequencies had a larger drop-off in amplitude (i.e. much lower contrast and therefore visual salience). The target object luminance was then adjusted to match the mean background luminance. Set upon the floor, at the far left- and right-hand sides of the scene, were two transparent box meshes ($210 \times 650$ pixels) that would act as 'spawn' (appearance) areas and govern the subsequent prey item movement vectors.

Above this ground activity, a camera item was positioned, which would provide the perspective for each trial. The camera item was rotated 90° to the floor component and had equalized RGB values, creating a monochrome bird's-eye view of the pebble backdrop. At the highest point of the scene was a directional light source and a skylight. Between the camera and lighting systems was a plane (static mesh) that would serve as our 'water surface' and be responsible for creating the water caustic effect. The material for this plane comprised a flipbook that contained a series of frames. The necessary frames were created using the free 'Caustics Generator' (Dual Heights, www.dualheights.se/caustics) and pieced together in a single square image, tiled as in a storyboard with the first frame located in the top left corner and the last frame in the bottom right (GlueIt, www.github.com/Kavex/GlueIT). This tiled image was then edited in GIMP2 (GIMP, www.gimp.org): first converted to monochrome, then the black–white contrast was increased and finally white pixels were converted to the alpha (transparency) channel. The result was an image that was transparent in only the regions that corresponded to the caustic network, which could then be read in sequence by the flipbook tool. As the flipbook reads the tiled image in sequence (at any given frequency, from left to right and top to bottom), the visualized material of the plane subtly changes accordingly. Paired with the scenes' directional light passing through the newly transparent regions of the plane, the effect is a caustic flicker projected upon the substrate. For treatments involving static caustics, the material used for the plane component was simply held as the first frame in the sequence. The sharpness of caustic boundary could be adjusted by altering the height (z-axis) of the caustics plane relative to the floor plane in UNREAL ENGINE 4: caustic plane heights used for sharp and diffuse treatment groups were fixed at $z = 3000$ and 9000 respectively. The scale of simulated caustics could be adjusted by using different depth settings in the 'Caustics Generator'. The difference in scale roughly translated to a magnitude of 10: the area of an individual prey item covered 30% of the total area within fine-scale caustic shade, while a prey item in coarse-scale caustic shade covered 3%.

Screen recordings (60 s) of each simulated scene running in UNREAL ENGINE 4 were made (via Bandicam, www.bandicam.com) to create an external bank of stimulus videos. All stimulus videos were presented on an iPad Air I (Apple, CA, www.apple.com), which has an LCD capacitive touchscreen (disabled) with a resolution of 10.4 pixels mm$^{-1}$, screen size of $1536 \times 2048$ pixels and a refresh rate of 60 Hz. The iPad had waterproof housing (LifeProof, www.lifeproof.com) and was placed in a transparent waterproof bag (Overboard, www.over-board.co.uk) with its long dimension horizontal. Each scene was monochromatic, covered a screen area of $1680 \times 1020$ pixels and was viewed from a bird's-eye perspective. There were a total of eight treatment groups ($2^3$ factorial design), distinguished by the scale, sharpness and motion of caustics present: fine-scale diffuse moving, fine-scale diffuse static control, fine-scale sharp moving, fine-scale sharp static control, coarse-scale diffuse moving, coarse-scale diffuse static control, coarse-scale sharp moving, coarse-scale sharp static control. The four treatments in which the motion of caustics was static acted as experimental controls, allowing for equal spatial complexity across scenes with and without motion. Mean luminance of the scene varied with treatment background (fine sharp: 45 cd m$^{-2}$; coarse sharp: 42 cd m$^{-2}$; fine diffuse: 61 cd m$^{-2}$; coarse diffuse: 57 cd m$^{-2}$), measured directly from the screen with a Konica Minolta CS-100A photometer (Konica Minolta Sensing America, Inc., Ramsey, NJ, www.sensing.konicaminolta.us). The wavelength of the dominant frequency in the fine caustics was 111 pixels, that of the coarse-grained was 210. Taking the width of a caustic as the distance between the locations of most rapid change in luminance, the median width of the light bands of the sharp caustics was 32 pixels (inter-quartile range (IQR) 29–35) for the fine treatment and 31 pixels (IQR 28–43) for the coarse. The width in the diffuse treatments was 73 pixels (IQR 71–82) in the coarse treatment, and 84 (IQR 76–102) in the fine treatment. The slightly larger width in the latter was because of 'fusion' of some caustics when they were at the higher density.

The simulated prey item was a three-dimensional sphere with a matt surface and mean luminance equal to that of each treatment background. When viewed in the experiment, the prey item was a circle of diameter 44 pixels (1.2° visual angle) with apparent three-dimensional shape derived from the realistic projection of light to create shape-from-shading cues [34]. For a given trial, a prey item appeared at any random locus within one of two spawn regions and followed a linear movement vector towards another random locus in the opposite region. Movement was fixed at a speed of 249 pixel s$^{-1}$ (6.9 deg s$^{-1}$, when viewed from the trial divider) and the prey item continued to move back and forth along this vector for the duration of the trial. While a speed matched to that of real triggerfish prey would have been ideal, the choice was made difficult by their broad diet, which ranges from slow-moving molluscs to fast-moving fish. Instead, the movement speed was chosen through pilot testing to find a

speed that fish would readily respond to, and be capable of pecking at, in nearly all trials. Appearance location and subsequent movement vectors were random, picked from discrete uniform distributions using UNREAL ENGINE's random integer generator. Location regions were set such that prey items never left the viewed scene.

## (c) Training phase

A total of seven training stages were required to introduce each experimental aspect in turn. Each training stage included a number of sessions: a total of 10 trials were completed per session, with two sessions daily (morning and afternoon). First, fish were presented with a static prey item hand-drawn on to a white PVC feeding board, identical in dimensions to the simulated prey item and iPad respectively. The pieces of squid used as a reward were naturally adhesive and could be stuck onto the feeding board or iPad screen, as appropriate. Fish were initially encouraged to approach the feeding board and prey item by positioning some squid next to the prey item (five sessions). For fish that did not immediately approach, the board could be left in the tank for an extended period to allow food to be taken in their own time and to reduce the novelty of the board. Second, the prey item with accompanying squid was presented on an iPad displaying a white background (seven sessions). Third, fish approached and pecked the prey item without food present (six sessions), with fish being tweezer-fed a squid reward immediately after a successful peck. At this point fish could then progress on to attacking a moving prey item (seven sessions).

The final three stages of training introduced (i) the trial divider and camera for recording behaviour (six sessions), (ii) an example set of simulated caustics overlaid on a white background (six sessions), and (iii) the experimental scene substrate without caustics (four sessions). Fish moved on to the next training stage when they had completed at least four sessions and achieved at least an 80% cumulative success rate. Eleven fish met this criterion for all training stages and entered the experiment.

The procedure for the final training sessions remained the same throughout experimentation. The trial divider was positioned in the aquarium to restrict the fish to the non-iPad end of the aquarium, together with the camera which was set to record. The trial divider was a large section of white plastic that, when placed in the tank (25 cm from the iPad), isolated the fish from the iPad and blocked the view of the camera. The camera, an Akaso V50 Pro (Akaso, www.akaso.net; 4 K resolution, 30 fps and 170° viewing angle), was housed in a waterproof case and attached with a suction clip onto the left wall of the aquarium, 10 cm from the non-iPad aquarium wall. The video files of a given treatment were queued and shuffled on the iPad. The divider was lifted (trial start) to allow the fish to find the prey item (see electronic supplementary material, movies S2 and S3). Upon a successful peck (trial end), fish were rewarded with the food item and the trial divider refitted. The next video file was loaded, and the process repeated. *Post hoc* video analysis was used to measure the time taken to accomplish the task ('Attack Latency', trial start to trial end).

Both feeding board and iPad were presented in the same way: lowered in on a modified hand net to the far end of the aquarium, perpendicular to the base. Though caustics in nature would differ in form if viewed from this perpendicular angle [6], it minimized the influence that angle of attack had on the ability to see the target: most fish would approach a face on target in a more uniform manner, whereas controls of entry would need to be installed if approached top-down. Water input was also shut off for every feed and training session (i) to avoid washing the squid off the board, and (ii) to act as a cue for the fish that food was imminent—most fish would leave their homes at this cue. A detailed overview of the aquarium set-up (electronic supplementary material, figure S1) is included within the electronic supplementary material.

## (d) Experimental protocol

All fish were tested twice per day (morning testing period and afternoon testing period) for four days. Each testing period involved the principal investigator presenting a fish with 10 trials of a given treatment whereby, upon completion, the next fish would be presented 10 trials of a different treatment, and so on. The end of the testing period was signified when all fish had completed 10 trials of their given treatment. The order of treatments presented across the four days was different for all fish. After four days, this process was repeated, staggering the treatment order by one to minimize any influence that morning versus afternoon testing periods may have upon motivation and satiation levels. A final repeat (using testing periods with five trials each) ensured that each fish had completed a total of 25 trials per treatment. Throughout the experimental phase, fish pecked the prey item within a range of 0.6 and 52 s (median Attack Latency 3.2 s, interquartile range 4.4 s) after presenting the stimulus (electronic supplementary material, figure S2). Trials in which the fish did not peck (four out of 2200; 0.2%) are excluded from the analysis.

## (e) Statistical analysis

All statistical analyses were performed in R v. 3.3.2 (R Foundation for Statistical Computing, www.R-project.org) and used linear mixed models (function lmer in the lme4 package [35]). The response variable was Attack Latency (log transformed) with Gaussian error and identity link functions. The transformation was necessary to normalize residuals in the face of skew in the raw time data. The primary model included the fixed effects caustic motion (static versus dynamic), caustic scale (fine versus coarse) and caustic sharpness (diffuse versus sharp), the three- and two-way interactions and the random effect of fish ID (to account for variation in task ability; electronic supplementary material, figure S3). Initially, the change in deviance between the model with and without the predictors of interest was tested against a $\chi^2$-distribution with degrees of freedom equal to the difference in degrees of freedom between the models. A secondary model included the fixed effect treatment and the random effect of fish ID.

## 3. Results

Our data demonstrate that the detection of moving prey items by Picasso triggerfish is significantly disrupted by the presence of dynamic illumination (figure 2). This effect can be directly attributed to changes in the three features of water caustics which were manipulated: fish were significantly slower to attack the prey item when caustics were moving ($\chi^2 = 290$, d.f. = 1, $p < 0.001$), when caustic boundaries were sharper ($\chi^2 = 27.6$, d.f. = 1, $p < 0.001$) and when the scale of caustic shade was fine ($\chi^2 = 15.1$, d.f. = 1, $p < 0.001$). Fish were slower to attack prey items that were presented among water caustics that were both fine in scale and sharp, a pattern more likely in the shallowest waters [6]. The fastest attack latencies arose when the presented caustics were static, irrespective of scale and sharpness, with these four (control) treatments indicating a baseline for triggerfish responses for this task. There were no statistically significant interactions between any factors (three-way interaction: $\chi^2 = 0.67$, d.f. = 10, $p = 0.41$; two-way interactions: scale × sharpness: $\chi^2 = 2.68$, d.f. = 1, $p = 0.10$; scale × motion: $\chi^2 = 0.05$, d.f. = 1, $p = 0.82$; motion × sharpness: $\chi^2 = 1.78$, d.f. = 1, $p = 0.18$).

## 4. Discussion

We tested the hypothesis that water caustic flicker (like other forms of dynamic illumination) masks the motion of a chosen

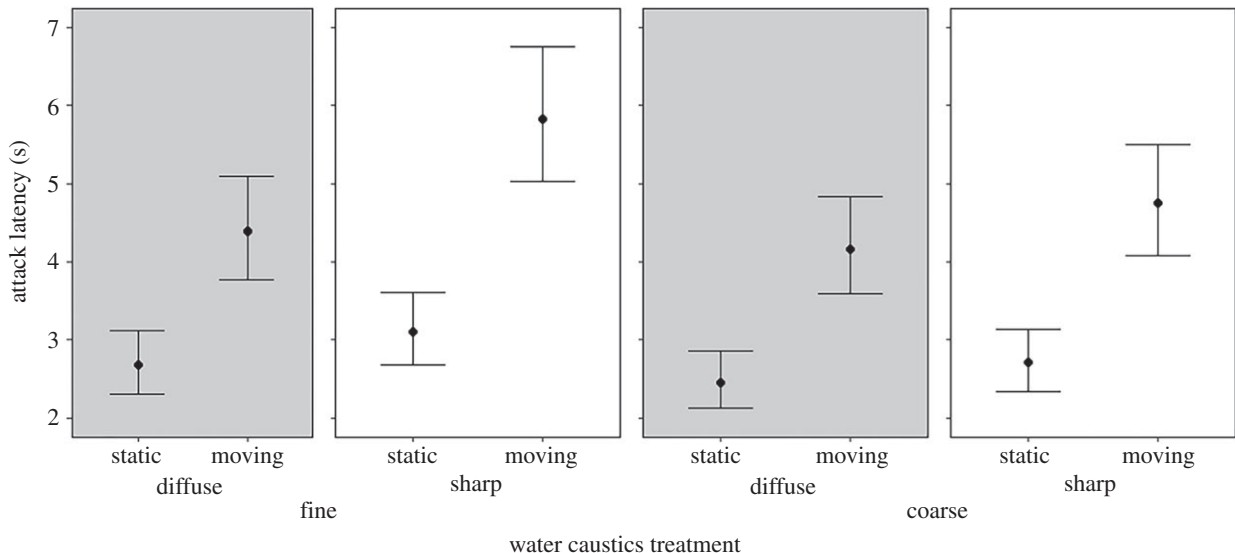

**Figure 2.** Mean attack latency of fish across the eight water caustics treatments. The treatments are initially divided by scale (fine versus coarse) then by sharpness (diffuse versus sharp) and motion (static versus moving). Error bars for attack latency indicate 95% confidence intervals derived from the linear mixed models.

target. Our data show that moving prey items are more difficult to detect and capture when the illumination in the surrounding scene is moving, rather than when equally spatially complex but static. As demonstrated by Matchette *et al.* [24], a motion signal can be masked to some degree if it falls within the distribution of motion noise [23], provided here by flickering water caustics.

While motion remained the most influential feature of water caustics tested, both the scale of caustic shade and the sharpness of caustic boundary were also important features; water caustics that were fine in scale and sharp in edge definition induced longer attack latencies (figure 2). It is the distance between the 'lens' of the waves and the substrate on to which light projects that determines how large and diffuse the caustics are. So, one inference might be that shallow habitats, where caustics will be fine-grained and sharply defined, may be key locations for visual disruption. However, because we used only one target size and one background type, this finding can be explained in several other ways. First, in the context of signal-to-noise ratio, the edge definition and size of the target matched more closely the structure of the caustics in the fine and sharp treatments rather than their (diffuse and coarse) counterparts. The target had a diameter of 44 pixels and the width of the light bands of the sharp caustics was 32 pixels, while that of the coarse caustics was 73–84 pixels and, in the diffuse treatments, less well defined. Although untested, one may therefore expect, for example, larger prey items to induce longer attack latencies when moving among coarse-scale water caustics than the same prey items within fine-scale caustics. Alternatively, the slower attack latencies associated with fine and sharp water caustic treatments may be due to a greater spatial complexity of such treatments, irrespective of the prey item size or definition. Whether it is the similarity of target and caustic size, or caustic stricture *per se*, or indeed an interaction of both with background characteristics, remain important directions for further study. While we expect the detectability of a moving target to be largely unaffected by the similarity of the target to the background [12], there could still be an interaction between background and caustic structure, and in turn with target characteristics, that affects detectability; this remains to be tested. It is also

important to note here that the differences in overall scene luminance between treatments may also underlie some of the observed effects. Therefore, while the testing of water caustic scale and sharpness proved both interesting and informative, further investigation is necessary to draw firm conclusions.

Our findings are the first to indicate how water caustics may influence the perception and behaviour of wild organisms: here, disrupting prey detection in Picasso triggerfish. This is despite triggerfish inhabiting the shallow habitats in which this rapidly changing illumination is most prevalent, living and feeding along the substrate for the entirety of their lives [25,26]. This highlights that, even with exposure to such visual noise, there are still limitations to their temporal vision. This contrasts with the situation for pelagic fish, where it has been suggested that caustics may be beneficial for prey detection. This untested hypothesis was based on the observation that the maximal temporal and spatial contrast sensitivities of many epipelagic marine organisms fall within the typical range of water caustic flicker [6,11,36]. This might facilitate the detection of reflective objects that are subsequently illuminated in midwater, particularly near the surface where flicker is most acute [6,11,36]. Therefore, the effects, positive or negative, of caustics may vary with ecology, a topic worthy of investigation.

We also predict that three-dimensionality of the substrate would accentuate this masking effect by increasing spatial complexity [6]: caustics projected upon rocks, foliage and coral are elongated and distorted. Indeed, for limbed or convoluted three-dimensional structures, such as mangroves and branched corals, the caustic flicker will illuminate several features concurrently, which may make estimations of a scene's depth or the spatial positioning of a target more difficult. Moreover, the variation of signal that arises from water caustic flicker is likely to affect not only the perceived motion of a specific prey item, but also feature binding of individuals [37,38], the perception of group movement [39], the type of movement an organism exhibits, and colour discrimination [40]; the latter, together with the effect upon features and movements, imply possible negative consequences of water caustics for signalling.

While our study remains an informative first step towards quantifying the influence of water caustics (and its visual

components) upon prey detection in wild organisms, there are some methodological caveats to consider that can direct future study. First, isolating the mechanisms by which different types of caustic are more or less deleterious: is a match in spatial characteristics to the target all-important (i.e. signal-to-noise ratio), or does spatial complexity *per se* have a role? Secondly, stimulus scenes were limited to monochrome: a decision based largely on the most obvious feature of caustics being their extreme luminance modulation, and motion detection representing a largely achromatic tasks for several animal groups (including fish) [27,29]. Yet, we recognize that many marine organisms are tri- or tetrachromatic, inhabit habitats that are particularly colourful, and caustics have coloured fringes. Our findings therefore represent a useful platform from which to extend the investigation, introducing aspects of colour to both the scene and the prey item and assess the subsequent efficacy of visual tasks beyond prey detection. Lastly, while the vertical placement of the iPad represented an effort to minimize variation in the perception of the scene with angle of approach, this method does not fully eradicate such variation. In future, the use of a centralized 'doorway', through which the fish must pass to reach the screen, may be a more appropriate method to control viewing conditions.

Overall, the motion and, to some extent, the form of water caustics appear to play an important role in visual foraging tasks and have a direct impact upon foraging efficiency. We predict, therefore, that water caustics (as with other forms of dynamic illumination) should influence decisions about habitat choice and foraging by both prey and predators in the wild.

**Ethics.** Fish were collected under a Great Barrier Reef Marine Park Authority Permit no. G16/38497 and Queensland General Fisheries Permit 183990. All procedures were approved by the Animal Welfare and Ethical Review Body of the University of Bristol (UIN/UB/18/084) and the Animal Ethics Committee at the University of Queensland (QBI/304/16).

**Data accessibility.** The datasets supporting this article have been uploaded as part of the electronic supplementary material.

**Authors' contributions.** All authors contributed to the conception and design of the study, as well as drafting the manuscript; S.R.M. programmed the stimuli in Unreal Engine 4, carried out fieldwork and participated in data analysis; K.L.C and N.J.M. provided fieldwork support; I.C.C. participated in data analysis. All gave final approval for publication.

**Competing interests.** We declare we have no competing interests.

**Funding.** This work was supported by S.R.M.'s CASE Studentship from the Engineering and Physical Sciences Research Council (EPSRC), UK, and QinetiQ plc.

**Acknowledgements.** We thank all the staff at the Lizard Island Research Station for their logistic support, as well as Erin Watson, Nadia Hamilton and Naomi Green for their expertise and assistance throughout the field-phase.

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
