## [Reviewer comments · Proceedings of the Royal Society B: Biological Sciences]

Review History

RSPB-2019-2453.R0 (Original submission)

Review form: Reviewer 1

Recommendation

Accept with minor revision (please list in comments)

Scientific importance: Is the manuscript an original and important contribution to its field?

Excellent

General interest: Is the paper of sufficient general interest?

Excellent

Quality of the paper: Is the overall quality of the paper suitable?

Excellent

Is the length of the paper justified?

Yes

Should the paper be seen by a specialist statistical reviewer?

Yes

Do you have any concerns about statistical analyses in this paper? If so, please specify them explicitly in your report.

No

It is a condition of publication that authors make their supporting data, code and materials available - either as supplementary material or hosted in an external repository. Please rate, if applicable, the supporting data on the following criteria.

Is it accessible?

Yes

Is it clear?

Yes

Is it adequate?

Yes

Do you have any ethical concerns with this paper?

No

Comments to the Author

This paper presents a well-thought out and executed study testing the effects of dynamic underwater caustics on a fish's ability to forage. The authors present sufficient detail of their experimental protocol for researchers to adequately reproduce the experiment, either with the same species or extend it to test the role of underwater caustics in additional species. This study will impact visual ecology studies seeking to understand the environmental pressures that give rise to animal vision and camouflage underwater. I have two major concerns: 1. The n-values are not reported for the number of fish tested nor the number of trials accomplished per fish. 2. The figures could use a bit of work to elevate them to the quality of the paper as a whole. In figure 1, a diagram of the experimental set up or a photograph of a trigger fish in the experimental arena would be more informative than a nice photo of a triggerfish. Figure 2 is difficult to visually interpret. To a naïve reader the number of experimental treatments in the study take a minute to grasp in the way presented, both in the figure and the text. This can be clarified in Figure 2 by modifying the organization of the x axis labels into a bracket-like system, as well as by separating the static/moving treatment pairs with some light gray shading. I provide a sketch with the minor comments related to what I mean.

Review form: Reviewer 2

Recommendation

Major revision is needed (please make suggestions in comments)

Scientific importance: Is the manuscript an original and important contribution to its field?

Excellent

General interest: Is the paper of sufficient general interest?

Good

Quality of the paper: Is the overall quality of the paper suitable?

Good

Is the length of the paper justified?

No

Should the paper be seen by a specialist statistical reviewer?

No

Do you have any concerns about statistical analyses in this paper? If so, please specify them explicitly in your report.

No

It is a condition of publication that authors make their supporting data, code and materials available - either as supplementary material or hosted in an external repository. Please rate, if applicable, the supporting data on the following criteria.

Is it accessible?

Yes

Is it clear?

Yes

Is it adequate?

No

Do you have any ethical concerns with this paper?

No

Comments to the Author

The authors present a behavioral study testing the effects of variable illumination given by water caustics on the salience of prey movement. Using the Picasso triggerfish as the viewer, the authors parameterize water caustics by sharpness, scale, and movement, then test the effects of each parameter on prey detection latency. The results of this experiment indicate that moving prey is more difficult to detect when illumination pattern in the surrounding scene is moving rather than when it is static. This finding is akin to our understanding of similar disruptive visual elements in terrestrial environments (such as windblown vegetation). The concept of this study is an important contribution to the visual ecology literature and the results are interesting, however I have several concerns about the execution and conclusions of this study that remain to be addressed.

Primary concerns:

The study includes eight different treatments in the experiment, testing coarse vs. fine, sharp vs diffuse, and moving vs static water caustics on prey detectability. It remains unclear why a 'substrate-only' treatment was omitted. Determining the effects of water caustics specifically, requires the variable of substrate complexity to be removed. At the very least, substrate cobble size should have been uniform across the visual field. Otherwise, the results may indicate an interaction of substrate cobble size to any of the parameters of water caustics. It is unclear why underlying substrate was not omitted and substituted for a uniform background for prey detection. The limitations of this study design need to be included in the discussion.

The study presents that fine (vs coarse) caustic scale significantly increases attack latency. However, an equally plausible interpretation is that prey size relative to caustic scale significantly alters attack latency. By presenting one prey size across different caustic scales, you cannot say that it's caustic scale itself that drives this pattern rather than the size of prey relative to the scale. The authors mention this relationship in the methods (Ln 122) however, the relevance of this point to the study findings is missing from the discussion.

In lines 118-124, the authors discuss visual adaptations of marine organisms to caustic flicker and that such adaptations facilitate target detection in pelagic and surface water environments. After, they use their results to argue that such visual adaptations must hinder target detection when

viewed against substrate. This would be a reasonable suggestion if the study model had midwater/pelagic visual adaptation. The triggerfish, however, is a benthic dwelling and substrate feeding organism and likely has visual capacities suited to this environment. The authors must clarify why their findings in triggerfish would support their claim.

At line 177, the authors describe that the iPad visualization of the substrate/caustics was presented at a perpendicular angle (90°?) to the aquarium floor, claiming that this artificial presentation accounts for angle-of-approach discrepancies that would be given by a benthic presentation. However, unless fish were trained to approach the board from a singular position, these same angle issues – and other issues related to activation of retinal region and angle of approach to prey – would still be present. Again, the limitations of this design should be included in the discussion.

Line 164 – It's stated that fish moved on to subsequent training/experimental phases when they achieved an 80% cumulative success rate. Because the study is dependent on attack latency, a more appropriate metric of behavioral readiness would have been stabilization of attack latency.

The plotting of 95% CI in Figure 2 is insufficient. It is acceptable to supplement the findings of linear modeling with frequentist statistics in order to indicate relationships between groups. Significance between groups should be noted.

Ln 137 – It's unclear why mean luminance differences between treatments were permitted and not corrected. This could have been accomplished by manipulating the brightness of certain scene elements to achieve equal treatment luminance. While prey luminance was set to the average of scene luminance (and thus has relative standardization between each treatment), these differences in overall luminance could underlie differences in retinal activation that would support vision; and again, is a limitation that should be included in the discussion.

The discussion section presented is brief and incomplete. There is no discussion of the specific findings of the study. For instance, from our knowledge in the literature or your own interpretation, why were sharp-edged and fine scale caustics more disruptive to prey detection? Also, why is there no discussion of the findings in figure 2?

Other concerns:

Please justify why scenes were presented as monochromatic.

Supplemental plots are required for further data transparency. The authors tested 16 fish, over 25 trials, across 8 treatments. Please provide plots visualizing these data, for example, with Fish ID on the x-axis with attack latency on the y axis (for each treatment). This will give the audience an indication of variability between fishes.

Did any fish, in any trial, not attack/detect the prey? If so, were those trials omitted? If so, why were they omitted? No attack may mean that prey went undetected and is important to include in your study findings. Please report the rate in which this happened.

In line 216, it's stated that prey are more difficult to detect when the 'surrounding scene is moving.' Unless, the substrate (i.e.,cobble) was also moving, then the scene wasn't moving just the illumination pattern over the scene.

Was the speed and size of the prey item provided biologically relevant to the Picasso Triggerfish? The authors need to better justify the details of the study.

Why is the abstract missing information about the results of the study, conclusions, and implications?

After initial presentation, either the common name or scientific name of the triggerfish should be presented throughout.

Triggerfish were said to be 65-130mm – what life stage is this (juvenile, adult?) and how might that relate to the study?

Please note in the methods if Unreal Engine 4 is a tool, program, software, etc.

Please explain how luminance was measured for each treatment? Is this information provided by Unreal Engine 4 – if so, say so

Review form: Reviewer 3

Recommendation

Accept with minor revision (please list in comments)

Scientific importance: Is the manuscript an original and important contribution to its field?

Marginal

General interest: Is the paper of sufficient general interest?

Marginal

Quality of the paper: Is the overall quality of the paper suitable?

Good

Is the length of the paper justified?

Yes

Should the paper be seen by a specialist statistical reviewer?

No

Do you have any concerns about statistical analyses in this paper? If so, please specify them explicitly in your report.

No

It is a condition of publication that authors make their supporting data, code and materials available - either as supplementary material or hosted in an external repository. Please rate, if applicable, the supporting data on the following criteria.

Is it accessible?

Yes

Is it clear?

Yes

Is it adequate?

Yes

Do you have any ethical concerns with this paper?

No

Comments to the Author

1. Please clarify more why comparing to static flicker is better than comparing to slowly moving caustics (large waves) to fast moving caustics (small waves). Seems to me that this comparison more natural.
2. In the test scenes shown on the iPad, I can see that they are black and white. However, in nature, does color has any biological significance for the fish to deal with flicker? Did you test any color scenes? If not why?
3. What about non-moving prey?

Decision letter (RSPB-2019-2453.R0)

20-Dec-2019

Dear Mr Matchette:

Your manuscript has now been peer reviewed and the reviews have been assessed by an Associate Editor. The reviewers' comments (not including confidential comments to the Editor) and the comments from the Associate Editor are included at the end of this email for your reference. As you will see, the reviewers and the Editors have raised some concerns with your manuscript and we would like to invite you to revise your manuscript to address them.

Research ethics:

Use of animals and field studies:

If your study uses animals please include details in the methods section of any approval and licences given to carry out the study and include full details of how animal welfare standards

were ensured. Field studies should be conducted in accordance with local legislation; please include details of the appropriate permission and licences that you obtained to carry out the field work.

If you wish to submit your data to Dryad (<http://datadryad.org/>) and have not already done so you can submit your data via this link [http://datadryad.org/submit?journalID=RSPB&manu=\(Document not available\)](http://datadryad.org/submit?journalID=RSPB&manu=(Document%20not%20available)), which will take you to your unique entry in the Dryad repository.

Please submit a copy of your revised paper within three weeks. If we do not hear from you within this time your manuscript will be rejected. If you are unable to meet this deadline please let us know as soon as possible, as we may be able to grant a short extension.

Best wishes,
Dr Daniel Costa
<mailto:proceedingsb@royalsociety.org>

Associate Editor
Board Member: 1
Comments to Author:

This manuscript explores whether prey detection by a visually-guided predator is disrupted by dynamic illumination. The authors present results from experiments with individual Picasso triggerfish that were trained to find and attack a moving prey item within simulated scenes with

varying forms of water caustics and if this is relative to specific visual features of water caustics: the scale of caustic shade and the sharpness of caustic boundary.

Introduction

The opening paragraph of the introduction includes a lot of technical details which, to the broader audience, needs some more unpacking. Line 34 when what is projected onto a 3D object? When you move onto projections onto planes, should this be a new sentence? Line 38 explain the meaning of reticulate. Line 40 Has this ever been tested, or are you the first?

line 41-48 I'd be tempted to start your whole introduction with this broader statement and context.

line 51 make the latter clearer

It might be a good idea to spell out the predictions in the introduction to help guide the reader through the results and to structure the discussion.

Experimental protocol

Make it clearer that all fish are presented with all eight treatments. It's also not clear why fish had 25 trials when it's stated earlier they have 10 trials per block and two blocks per day for four days - 80 trials. five extra trials is 15 - so I'm missing something here to help me understand the design fully.

Discussion

The discussion is very short. It refers to a hypothesis, but this was not stated in the introduction.

Reviewer two raises a number of important queries about the design of the experiment, e.g., the lack of a 'substrate-only' treatment, and also the prey size relative to caustic scale. The limitations of this study design, and these alternative explanations for the results need to be included in the discussion.

Reviewer(s)' Comments to Author:

Referee: 1

Comments to the Author(s)

This paper presents a well-thought out and executed study testing the effects of dynamic underwater caustics on a fish's ability to forage. The authors present sufficient detail of their experimental protocol for researchers to adequately reproduce the experiment, either with the same species or extend it to test the role of underwater caustics in additional species. This study will impact visual ecology studies seeking to understand the environmental pressures that give rise to animal vision and camouflage underwater. I have two major concerns: 1. The n-values are not reported for the number of fish tested nor the number of trials accomplished per fish. 2. The figures could use a bit of work to elevate them to the quality of the paper as a whole. In figure 1, a diagram of the experimental set up or a photograph of a trigger fish in the experimental arena would be more informative than a nice photo of a triggerfish. Figure 2 is difficult to visually interpret. To a naïve reader the number of experimental treatments in the study take a minute to grasp in the way presented, both in the figure and the text. This can be clarified in Figure 2 by modifying the organization of the x axis labels into a bracket-like system, as well as by separating the static/moving treatment pairs with some light gray shading. I provide a sketch with the minor comments related to what I mean.

Referee: 2

Comments to the Author(s)

The authors present a behavioral study testing the effects of variable illumination given by water caustics on the salience of prey movement. Using the Picasso triggerfish as the viewer, the authors parameterize water caustics by sharpness, scale, and movement, then test the effects of

each parameter on prey detection latency. The results of this experiment indicate that moving prey is more difficult to detect when illumination pattern in the surrounding scene is moving rather than when it is static. This finding is akin to our understanding of similar disruptive visual elements in terrestrial environments (such as windblown vegetation). The concept of this study is an important contribution to the visual ecology literature and the results are interesting, however I have several concerns about the execution and conclusions of this study that remain to be addressed.

Primary concerns:

The study includes eight different treatments in the experiment, testing coarse vs. fine, sharp vs diffuse, and moving vs static water caustics on prey detectability. It remains unclear why a 'substrate-only' treatment was omitted. Determining the effects of water caustics specifically, requires the variable of substrate complexity to be removed. At the very least, substrate cobble size should have been uniform across the visual field. Otherwise, the results may indicate an interaction of substrate cobble size to any of the parameters of water caustics. It is unclear why underlying substrate was not omitted and substituted for a uniform background for prey detection. The limitations of this study design need to be included in the discussion.

The study presents that fine (vs coarse) caustic scale significantly increases attack latency. However, an equally plausible interpretation is that prey size relative to caustic scale significantly alters attack latency. By presenting one prey size across different caustic scales, you cannot say that it's caustic scale itself that drives this pattern rather than the size of prey relative to the scale. The authors mention this relationship in the methods (Ln 122) however, the relevance of this point to the study findings is missing from the discussion.

In lines 118-124, the authors discuss visual adaptations of marine organisms to caustic flicker and that such adaptations facilitate target detection in pelagic and surface water environments. After, they use their results to argue that such visual adaptations must hinder target detection when viewed against substrate. This would be a reasonable suggestion if the study model had midwater/pelagic visual adaptation. The triggerfish, however, is a benthic dwelling and substrate feeding organism and likely has visual capacities suited to this environment. The authors must clarify why their findings in triggerfish would support their claim.

At line 177, the authors describe that the iPad visualization of the substrate/caustics was presented at a perpendicular angle (90°?) to the aquarium floor, claiming that this artificial presentation accounts for angle-of-approach discrepancies that would be given by a benthic presentation. However, unless fish were trained to approach the board from a singular position, these same angle issues – and other issues related to activation of retinal region and angle of approach to prey – would still be present. Again, the limitations of this design should be included in the discussion.

Line 164 – It's stated that fish moved on to subsequent training/experimental phases when they achieved an 80% cumulative success rate. Because the study is dependent on attack latency, a more appropriate metric of behavioral readiness would have been stabilization of attack latency.

The plotting of 95% CI in Figure 2 is insufficient. It is acceptable to supplement the findings of linear modeling with frequentist statistics in order to indicate relationships between groups. Significance between groups should be noted.

Ln 137 – It's unclear why mean luminance differences between treatments were permitted and not corrected. This could have been accomplished by manipulating the brightness of certain scene elements to achieve equal treatment luminance. While prey luminance was set to the average of scene luminance (and thus has relative standardization between each treatment), these differences in overall luminance could underlie differences in retinal activation that would support vision; and again, is a limitation that should be included in the discussion.

The discussion section presented is brief and incomplete. There is no discussion of the specific findings of the study. For instance, from our knowledge in the literature or your own interpretation, why were sharp-edged and fine scale caustics more disruptive to prey detection? Also, why is there no discussion of the findings in figure 2?

Other concerns:

Please justify why scenes were presented as monochromatic.

Supplemental plots are required for further data transparency. The authors tested 16 fish, over 25 trials, across 8 treatments. Please provide plots visualizing these data, for example, with Fish ID on the x-axis with attack latency on the y axis (for each treatment). This will give the audience an indication of variability between fishes.

Did any fish, in any trial, not attack/detect the prey? If so, were those trials omitted? If so, why were they omitted? No attack may mean that prey went undetected and is important to include in your study findings. Please report the rate in which this happened.

In line 216, it's stated that prey are more difficult to detect when the 'surrounding scene is moving.' Unless, the substrate (i.e.,cobble) was also moving, then the scene wasn't moving just the illumination pattern over the scene.

Was the speed and size of the prey item provided biologically relevant to the Picasso Triggerfish? The authors need to better justify the details of the study.

Why is the abstract missing information about the results of the study, conclusions, and implications?

After initial presentation, either the common name or scientific name of the triggerfish should be presented throughout.

Triggerfish were said to be 65-130mm – what life stage is this (juvenile, adult?) and how might that relate to the study?

Please note in the methods if Unreal Engine 4 is a tool, program, software, etc.

Please explain how luminance was measured for each treatment? Is this information provided by Unreal Engine 4 – if so, say so

Referee: 3

Comments to the Author(s)

1. Please clarify more why comparing to static flicker is better than comparing to slowly moving caustics (large waves) to fast moving caustics (small waves). Seems to me that this comparison more natural.
2. In the test scenes shown on the iPad, I can see that they are black and white. However, in nature, does color has any biological significance for the fish to deal with flicker? Did you test any color scenes? If not why?
3. What about non-moving prey?

Author's Response to Decision Letter for (RSPB-2019-2453.R0)

See Appendix A.

RSPB-2019-2453.R1 (Revision)

Review form: Reviewer 1

Recommendation

Accept as is

Scientific importance: Is the manuscript an original and important contribution to its field?

Excellent

General interest: Is the paper of sufficient general interest?

Excellent

Quality of the paper: Is the overall quality of the paper suitable?

Excellent

Is the length of the paper justified?

Yes

Should the paper be seen by a specialist statistical reviewer?

No

Do you have any concerns about statistical analyses in this paper? If so, please specify them explicitly in your report.

No

It is a condition of publication that authors make their supporting data, code and materials available - either as supplementary material or hosted in an external repository. Please rate, if applicable, the supporting data on the following criteria.

Is it accessible?

Yes

Is it clear?

Yes

Is it adequate?

Yes

Do you have any ethical concerns with this paper?

No

Comments to the Author

All corrections made to the manuscript are acceptable and I am happy with this paper as is.

Review form: Reviewer 2

Recommendation

Accept as is

Scientific importance: Is the manuscript an original and important contribution to its field?
Excellent

General interest: Is the paper of sufficient general interest?
Good

Quality of the paper: Is the overall quality of the paper suitable?
Excellent

Is the length of the paper justified?
Yes

Should the paper be seen by a specialist statistical reviewer?
No

Do you have any concerns about statistical analyses in this paper? If so, please specify them explicitly in your report.
No

It is a condition of publication that authors make their supporting data, code and materials available - either as supplementary material or hosted in an external repository. Please rate, if applicable, the supporting data on the following criteria.

Is it accessible?
Yes

Is it clear?
Yes

Is it adequate?
Yes

Do you have any ethical concerns with this paper?
No

Comments to the Author

I appreciate the author's efforts to address my concerns. This was an ambitious study; one that I think is an important contribution to the literature.

I have no further concerns or revisions to suggest.

Decision letter (RSPB-2019-2453.R1)

09-Mar-2020

Dear Mr Matchette

I am pleased to inform you that your manuscript entitled "Underwater caustics disrupt prey detection by a reef fish" has been accepted for publication in Proceedings B.

You can expect to receive a proof of your article from our Production office in due course, please check your spam filter if you do not receive it. PLEASE NOTE: you will be given the exact page

length of your paper which may be different from the estimation from Editorial and you may be asked to reduce your paper if it goes over the 10 page limit.

Open Access

Your article has been estimated as being 8 pages long. Our Production Office will be able to confirm the exact length at proof stage.

Paper charges

Sincerely,

Dr Daniel Costa

Appendix A

Responses to referees' and associate editor's comments

Original comments in black italics; our responses in plain red font.

Quoted line numbers relate to the manuscript when change tracking is set to 'simple markup'.

Associate Editor

Board Member: 1

Comments to Author:

This manuscript explores whether prey detection by a visually-guided predator is disrupted by dynamic illumination. The authors present results from experiments with individual Picasso triggerfish that were trained to find and attack a moving prey item within simulated scenes with varying forms of water caustics and if this is relative to specific visual features of water caustics: the scale of caustic shade and the sharpness of caustic boundary.

Introduction

The opening paragraph of the introduction includes a lot of technical details which, to the broader audience, needs some more unpacking. Line 34 when what is projected onto a 3D object?

We have adjusted the opening paragraph and reduced the 'jargon density'. We have added an explanation of why caustics have been implicated in the evolution of colour vision, rather than leaving it to the reader to hunt down that reference. It is the caustics that we were referring to being projected onto a 3D object; we have added that, and this has been moved to the start of paragraph two.

When you move onto projections onto planes, should this be a new sentence?

Adjusted accordingly.

Line 38 explain the meaning of reticulate.

We have deleted the jargon 'reticulate' because it is unnecessary; hopefully the figure makes it clear what we are referring to when describing the patterns.

Line 40 Has this ever been tested, or are you the first?

Here, we are not testing whether certain fish markings can act as effective camouflage amongst water caustics, but we are testing whether the dynamism of water caustics can effectively reduce the saliency of motion signals. To our knowledge, we are the first to test this, but don't feel we need to say so in the manuscript.

line 41-48 I'd be tempted to start your whole introduction with this broader statement and context.

We agree; see above regarding the restructuring.

line 51 make the latter clearer

Adjusted accordingly.

It might be a good idea to spell out the predictions in the introduction to help guide the reader through the results and to structure the discussion.

Good suggestion. We have added "...hypothesising that water caustic flicker will mask the motion of a target prey item" and, later, "we hypothesise that water caustics with sharp boundaries and fine shade scale - which represent those most acute in shallower waters - will induce the greatest attack latencies upon the current prey item"

Experimental protocol

Make it clearer that all fish are presented with all eight treatments. It's also not clear why fish had 25 trials when it's stated earlier they have 10 trials per block and two blocks per day for four days - 80 trials. five extra trials is 15 - so i'm missing something here to help me understand the design fully.

This has hopefully been made clearer (see Referee #1). We had failed to make it clear that fish completed 25 trials for each treatment, which boiled down to two blocks of 10 and one block of 5.

Discussion

The discussion is very short. It refers to a hypothesis, but this was not stated in the introduction.

See our response, above, to the need to spell out the predictions.

Reviewer two raises a number of important queries about the design of the experiment, e.g., the lack of a 'substrate-only' treatment, and also the prey size relative to caustic scale. The limitations of this study design, and these alternative explanations for the results need to be included in the discussion.

Adjusted accordingly (see responses to Referee #2)

Reviewer(s)' Comments to Author:

Referee: 1

Comments to the Author(s)

This paper presents a well-thought out and executed study testing the effects of dynamic underwater caustics on a fish's ability to forage. The authors present sufficient detail of their experimental protocol for researchers to adequately reproduce the experiment, either with the same species or extend it to test the role of underwater caustics in additional species. This study will impact visual ecology studies seeking to understand the environmental pressures that give rise to animal vision and camouflage underwater.

I have two major concerns:

1. The n-values are not reported for the number of fish tested nor the number of trials accomplished per fish.

Adjusted accordingly. The number of original (16) and analysed (11) fish has been included (line 75 and line 193 respectively, as well as Fig. 2) and the number of trials (25 for each of the 8 treatments) has been made clearer (lines 213 to 220), reading as follows: "All fish were tested twice per day (morning testing period and afternoon testing period) for four days. Each testing period involved the principal investigator presenting a fish with 10 trials of a given treatment whereby, upon completion, the next fish would be presented 10 trials of a different treatment, and so on. The end of the testing period was signified when all fish had completed 10 trials of their given treatment. The order of treatments presented across the four days was different for all fish. After four days, this process was repeated, staggering the treatment order by one to minimise any influence that morning vs afternoon testing periods may have upon motivation and satiation levels. A final repeat (using testing periods with five trials each) ensured that each fish had completed a total of 25 trials per treatment."

2. The figures could use a bit of work to elevate them to the quality of the paper as a whole.

In figure 1, a diagram of the experimental set up or a photograph of a trigger fish in the experimental arena would be more informative than a nice photo of a triggerfish.

Adjusted accordingly. The photo has been replaced with a post-hoc trial frame and a diagram relating to the aquaria set-up has been included in the Supplementary Material (Fig.S1).

Figure 2 is difficult to visually interpret. To a naïve reader the number of experimental treatments in the study take a minute to grasp in the way presented, both in the figure and the text. This can be clarified in Figure 2 by modifying the organization of the x axis labels into a bracket-like system, as well as by separating the static/moving treatment pairs with some light gray shading. I provide a sketch with the minor comments related to what I mean.

Excellent suggestion, which we have implemented in a new figure 2.

Line 58. Feeding to feed

Adjusted accordingly.

Line 170. For the trials, was there a maximum time limit given to animals to peck the target? I inferred from the data that 100% of the animals pecked, eventually, but I would like to see the range of time periods in which it took an animal to accomplish the task, rather than just a report that 99% percent of the trials occurred within 1 min (line 190). Is the time to accomplish the task referred to as the Attack latency? It would be nice to see a figure showing these data.

Adjusted accordingly. A new figure is included within the supplementary material to show the range of attack times (Fig. S2). The term 'Attack Latency' is also introduced more clearly (line 201).

Line 191. What is IQR? Not defined.

Interquartile range – now defined when first used.

Line 203: Results – what were your final n values? # of animals? # of trials per animal? Did each individual animal vary its ability to spot the prey, or were statistics only performed on the pooled data set?

A total of 11 fish completed the experiment with each fish completing 25 trials per treatment (200 trials in total). This has been made clearer (line 192).

Each fish tested differed, to some extent, in its ability to complete the task (see Fig.S3). To account for this, Fish ID was included in the linear mixed model as a random effect.

Referee: 2

Comments to the Author(s)

The authors present a behavioral study testing the effects of variable illumination given by water caustics on the salience of prey movement. Using the Picasso triggerfish as the viewer, the authors parameterize water caustics by sharpness, scale, and movement, then test the effects of each parameter on prey detection latency. The results of this experiment indicate that moving prey is more difficult to detect when illumination pattern in the surrounding scene is moving rather than when it is static. This finding is akin to our understanding of similar disruptive visual elements in terrestrial environments (such as windblown vegetation). The concept of this study is an important contribution to the visual ecology literature and the results are interesting, however I have several concerns about the execution and conclusions of this study that remain to be addressed.

Primary concerns:

The study includes eight different treatments in the experiment, testing coarse vs. fine, sharp vs diffuse, and moving vs static water caustics on prey detectability. It remains unclear why a 'substrate-only' treatment was omitted. Determining the effects of water caustics specifically, requires the variable of substrate complexity to be removed. At the very least, substrate cobble size should have been uniform across the visual field. Otherwise, the results may indicate an interaction of substrate cobble size to any of the parameters of water caustics. It is unclear why underlying substrate was not omitted and substituted for a uniform background for prey detection. The limitations of this study design need to be included in the discussion.

The referee is correct that having single background type is certainly a limitation. The choice of the background was, in part, simply from a desire to have a more 'naturalistic' background. If we had used uniform grey, a different referee might question how our results would apply to natural, more varied, substrates. In either case, using one background type reduces the external validity of the results, but testing more background types would have increased the, already considerable, challenge of running fish through enough trials. We also felt that, based on previous work (Hall et al. 2013, and research in the vision science literature, cited in that paper), with a moving target, similarity or dissimilarity of the target to the background would have little, if any, effect. Motion breaks camouflage, even if the target matches the background perfectly when static. However, an interaction with the structure of the caustics is certainly possible. We have added to the discussion "While we expect the detectability of a moving target to be largely unaffected by the similarity of the target to the background [12], there could still be an interaction between background and caustic structure, and in turn with target characteristics, that affects detectability; this remains to be tested." We have also added details about the background's spectral characteristics to the Methods: "The background had no single dominant spatial frequency (i.e. no predominant pebble size), with a log-log plot of the amplitude of the spatial frequency against the frequency itself (down to a frequency equivalent to half the diameter of the target) having a slope of -0.91. Such a relationship is typical for many natural scenes, a slope of -1 being common [Tolhurst et al. 1992]. The amplitude-frequency relationship for objects smaller than about half the size of the target was much steeper. That is, higher spatial frequencies had a larger drop-off in amplitude (i.e. much lower contrast and therefore visual salience)." Where the reference is Tolhurst DJ, Tadmor Y, Chao T. 1992 Amplitude spectra of natural images. *Ophthalmic. Physiol. Opt.* **12**, 229-232.

The study presents that fine (vs coarse) caustic scale significantly increases attack latency. However, an equally plausible interpretation is that prey size relative to caustic scale significantly alters attack latency. By presenting one prey size across different caustic scales, you cannot say that it's caustic scale itself that drives this pattern rather than the size of prey relative to the scale. The authors mention this relationship in the methods (Ln 122) however, the relevance of this point to the study findings is missing from the discussion.

Excellent point. We have now included the information on spatial characteristics of the caustics, as well as the pebbly substrate (see above) in the Methods, as the basis for the discussion. The new information in the Methods is "The wavelength of the dominant frequency in the fine caustics was 111 pixels, that of the coarse-grained was 210. Taking the width of a caustic as the distance between the locations of most rapid change in luminance, the median width of the light bands of the sharp caustics was 32 pixels (inter-quartile range (IQR) 29 - 35) for the fine treatment and 31 pixels (IQR 28 - 43) for the coarse. The width in the diffuse treatments was 73 pixels (IQR 71 - 82) in the coarse treatment, and 84 (IQR 76-102) in the fine treatment. The slightly larger width in the latter was because of 'fusion' of some caustics when they were at the higher density." A new second paragraph in the Discussion addresses the possibilities raised by the referee. It reads "While motion remained the most influential feature of water caustics

tested, both the scale of caustic shade and the sharpness of caustic boundary were also important features: water caustics that were fine in scale and sharp in edge definition induced longer attack latencies (Fig.2). It is the distance between the ‘lens’ of the waves and the substrate on to which light projects that determines how large and diffuse the caustics are. So, one inference might be that shallow habitats (where such features are most common) may be key locations for visual disruption. However, because we used only one target size and one background type, this finding can be explained in several other ways. First, in the context of signal-to-noise ratio, the edge definition and size of the target matched more closely the structure of the caustics in the fine and sharp treatments rather than their (diffuse and coarse) counterparts. The target had a diameter of 44 pixels and the width of the light bands of the sharp caustics was 32 pixels, while that of the coarse caustics was 73 – 84 pixels and, in the diffuse treatments, less well defined. Although untested, one may therefore expect, for example, larger prey items to induce longer attack latencies when moving among coarse-scale water caustics than the same prey items within fine-scale. Alternatively, the slower attack latencies associated with fine and sharp water caustic treatments may be due to a greater spatial complexity of such treatments, irrespective of the prey item size or definition. Whether it is the similarity of target and caustic size, or caustic structure per se, or indeed an interaction of both with background characteristics, remain important directions for further study. It is also important to note here that the differences in overall scene luminance between treatments may also underly the visual disparity via differences in retinal activation. Therefore, while the testing of water caustic scale and sharpness proved both interesting and informative, further investigation is necessary to draw firm conclusions.”

In lines 118-124, the authors discuss visual adaptations of marine organisms to caustic flicker and that such adaptations facilitate target detection in pelagic and surface water environments. After, they use their results to argue that such visual adaptations must hinder target detection when viewed against substrate. This would be a reasonable suggestion if the study model had midwater/pelagic visual adaptation. The triggerfish, however, is a benthic dwelling and substrate feeding organism and likely has visual capacities suited to this environment. The authors must clarify why their findings in triggerfish would support their claim.

We are afraid that our attempt to incorporate suggestions of previous authors served only to confuse our main message. We were not proposing that caustics facilitate prey capture nor applying this idea to our study. As the referee says, triggerfish feed on the substrate and so what happens in the pelagic zone is irrelevant. It was McFarland and Loew who proposed this, in 1983, based on their subjective experience when diving and a proposed tight fit between the temporal characteristics of caustics and the critical flicker fusion frequency of pelagic fish. We know of no hard data that substantiate their idea, but we felt we should mention the hypothesis, precisely because it is in the opposite direction (caustics enhance prey detection) to the results we obtained (caustics impede prey detection). An effort to address this has been included within the modified Discussion (paragraph three), which now reads: “Our findings are the first to indicate how water caustics may influence the perception and behaviour of wild organisms: here, disrupting prey detection in Picasso triggerfish. This is despite triggerfish inhabiting the shallow habitats in which this rapidly changing illumination is most prevalent, living and feeding along the substrate for the entirety of their lives [25,26]. This highlights that, even with exposure to such visual noise, there are still limitations to their temporal vision. This contrasts with the situation for pelagic fish, where it has been suggested that caustics may be beneficial for prey detection. This untested hypothesis was based on the observation that the maximal temporal and spatial contrast sensitivities of many epipelagic marine organisms fall within the typical range of water caustic flicker [6,11,35]. This might facilitate the detection of reflective objects that are subsequently illuminated in midwater, particularly near the surface where flicker is most acute [6,11,35]. Therefore the effects, positive or negative, of caustics may vary with ecology, a topic worthy of investigation.”

At line 177, the authors describe that the iPad visualization of the substrate/caustics was presented at a perpendicular angle (90°?) to the aquarium floor, claiming that this artificial presentation accounts for angle-of-approach discrepancies that would be given by a benthic presentation. However, unless fish were trained to approach the board from a singular position, these same angle issues – and other issues related to activation of retinal region and angle of approach to prey – would still be present. Again, the limitations of this design should be included in the discussion.

From observations during training and trial videos, the preferred route by fish remained relatively central and high in the water, approaching the iPad at roughly the same angle. However, you are correct in highlighting that this isn’t fully controlled, and angle issues will remain. Paragraph five of the Discussion has this addition: “Lastly, while the vertical placement of the iPad represented an effort to minimise variation in the perception of the scene with angle of approach, this method does not fully eradicate such variation. In future, the use of a centralised ‘doorway’, through which the fish must pass to reach the screen, may be a more appropriate method to control viewing conditions.”.

Line 164 – It's stated that fish moved on to subsequent training/experimental phases when they achieved an 80% cumulative success rate. Because the study is dependent on attack latency, a more appropriate metric of behavioral readiness would have been stabilization of attack latency.

Thank you for the suggestion – we will consider this approach for future behavioural work.

The plotting of 95% CI in Figure 2 is insufficient. It is acceptable to supplement the findings of linear modeling with frequentist statistics in order to indicate relationships between groups. Significance between groups should be noted.

We feel that the analysis presented fairly represents the experimental design, which was a 2x2x2 factorial. To start to compare individual groups (e.g. with Tukey tests) starts to look like fishing, is less efficient (reduced statistical power) and not how we conceived the experiment. However, because the raw data are available on Dryad, other researchers will be free to reanalyse the data how they wish.

Ln 137 – It's unclear why mean luminance differences between treatments were permitted and not corrected. This could have been accomplished by manipulating the brightness of certain scene elements to achieve equal treatment luminance. While prey luminance was set to the average of scene luminance (and thus has relative standardization between each treatment), these differences in overall luminance could underlie differences in retinal activation that would support vision; and again, is a limitation that should be included in the discussion.

A valid point and these luminance differences are unfortunate – we only detected this after the experiment was completed and, the lead experimenter/author being UK based, we could not simply rerun the experiment. This point is now highlighted within the modified Discussion as follows: “It is also important to note here that the differences in overall scene luminance between treatments may also underlie some of the observed effects. Therefore, while the testing of water caustic scale and sharpness proved both interesting and informative, further investigation is necessary to draw firm conclusions.”

The discussion section presented is brief and incomplete. There is no discussion of the specific findings of the study. For instance, from our knowledge in the literature or your own interpretation, why were sharp-edged and fine scale caustics more disruptive to prey detection? Also, why is there no discussion of the findings in figure 2?

We apologise for the briefness of the discussion; the result of trying to get under the word limit. A clear discussion is of course more important than saved page charges, so we hope the expansion of the discussion in the ways described above (e.g. the spatial characteristics of the caustics and background relative to the target and how this might mediate the observed effects, the variation in mean luminance, the contrast with hypotheses about the effect of caustics in pelagic habitats) is an improvement.

Other concerns:

Please justify why scenes were presented as monochromatic.

This has been addressed in the Discussion (paragraph five) as follows: “Second, stimulus scenes were limited to monochrome: a decision based largely on the most obvious feature of caustics being their extreme luminance modulation, and motion detection representing a largely achromatic task for several animal groups (including fish) [27,29]. Yet, we recognise that many marine organisms are tri- or tetrachromatic, inhabit habitats that are particularly colourful, and caustics have coloured fringes. Our findings therefore represent a useful platform from which to extend the investigation, introducing aspects of colour to both the scene and the prey item, and assess the subsequent efficacy of visual tasks beyond prey detection.”

Supplemental plots are required for further data transparency. The authors tested 16 fish, over 25 trials, across 8 treatments. Please provide plots visualizing these data, for example, with Fish ID on the x-axis with attack latency on the y axis (for each treatment). This will give the audience an indication of variability between fishes.

Adjusted accordingly. A new figure is included within the supplementary material (Fig. S3) to show the mean attack latencies for each treatment across all fish. Remember, however, that the raw data are available and so readers can explore these data in further detail.

Did any fish, in any trial, not attack/detect the prey? If so, were those trials omitted? If so, why were they omitted? No attack may mean that prey went undetected and is important to include in your study findings. Please report the rate in which this happened.

Of the 2200 trials, only four trials (0.2%) included fish that failed to find/peck the prey item within the designated time limit (60 s). While we agree that it is important to report when stimuli remain undetected (which we have now done: “Trials in which the fish did not peck (four out of 2200; 0.2%) are excluded from the analysis”), we believe that this percentage is negligible and further discussion is unnecessary.

In line 216, it's stated that prey are more difficult to detect when the 'surrounding scene is moving.' Unless, the substrate (i.e., cobble) was also moving, then the scene wasn't moving just the illumination pattern over the scene.

Adjusted accordingly: “when the illumination in the surrounding scene is moving”.

Was the speed and size of the prey item provided biologically relevant to the Picasso Triggerfish? The authors need to better justify the details of the study.

A fair point, and we have added this to the Methods: “While a speed matched to that of real triggerfish prey would have been ideal, the choice was made difficult by their broad diet, which ranges from slow-moving molluscs to fast-moving fish. Instead, the movement speed was chosen through pilot testing to find a speed that fish would readily respond to, and be capable of pecking at in nearly all trials”.

Why is the abstract missing information about the results of the study, conclusions, and implications?

Adjusted accordingly.

After initial presentation, either the common name or scientific name of the triggerfish should be presented throughout.

Adjusted accordingly.

Triggerfish were said to be 65-130mm – what life stage is this (juvenile, adult?) and how might that relate to the study?

A note regarding life stage has been included as follows: “individuals were deemed to be subadults and adults, and displayed similar levels of motivation to peck at prey items of the size presented”.

Please note in the methods if Unreal Engine 4 is a tool, program, software, etc.

The text now says “the software Unreal Engine 4”.

Please explain how luminance was measured for each treatment? Is this information provided by Unreal Engine 4 – if so, say so

Adjusted accordingly. We have added the following text (line 152): “measured directly from the screen with a Konica Minolta CS-100A photometer (Konica Minolta Sensing America, Inc., Ramsey, NJ, USA; www.sensing.konicaminolta.us)”.

Referee: 3

Comments to the Author(s)

1. Please clarify more why comparing to static flicker is better than comparing to slowly moving caustics (large waves) to fast moving caustics (small waves). Seems to me that this comparison more natural.

While that would appear a more natural option, these controls are still inappropriate for the coastal habitats in question: larger waves are relatively uncommon in shallow areas (unless conditions are perfect) and smaller waves (that influence water caustic form) are uncommon in deeper waters. While the controls we have chosen are superficial, they are easier to implement and fully control for spatial complexity.

2. In the test scenes shown on the iPad, I can see that they are black and white. However, in nature, does color has any biological significance for the fish to deal with flicker? Did you test any color scenes? If not why?

This has been addressed – see our response to referee 2.

3. What about non-moving prey?

A good suggestion, especially given that slow-moving/static prey is included within the diet of Picasso triggerfish. With more time on Lizard Island, we might have examined the effect of target speed. However, previous work on fish by Ioannou & Krause (2009) as well as by us on humans and birds (Matchette et al., 2018, *Anim Behav*; Matchette et al., 2019, *Anim Behav*) has already demonstrated the effectiveness of remaining stationary; hence we expect stationary prey items to remain the hardest to spot, irrespective of the illumination complexity. The key here is to emphasise the circumstances within which motion signals, which typically increase saliency and can aid identification/capture, can be masked.